# Characterization of the Shape Anisotropy of Superparamagnetic Iron Oxide Nanoparticles during Thermal Decomposition

**DOI:** 10.3390/ma13092018

**Published:** 2020-04-25

**Authors:** Dimitri Vanhecke, Federica Crippa, Marco Lattuada, Sandor Balog, Barbara Rothen-Rutishauser, Alke Petri-Fink

**Affiliations:** 1Adolphe Merkle Institute, University of Fribourg, Chemin des Verdiers 4, CH-1700 Fribourg, Switzerland; 2Chemistry Department, University of Fribourg, Chemin du Musée 9, 1700 Fribourg, Switzerland

**Keywords:** anisotropy, SPIONs, electron tomography, thermal decomposition, stereology, preferred orientation

## Abstract

Magnetosomes are near-perfect intracellular magnetite nanocrystals found in magnetotactic bacteria. Their synthetic imitation, known as superparamagnetic iron oxide nanoparticles (SPIONs), have found applications in a variety of (nano)medicinal fields such as magnetic resonance imaging contrast agents, multimodal imaging and drug carriers. In order to perform these functions in medicine, shape and size control of the SPIONs is vital. We sampled SPIONs at ten-minutes intervals during the high-temperature thermal decomposition reaction. Their shape (sphericity and anisotropy) and geometric description (volume and surface area) were retrieved using three-dimensional imaging techniques, which allowed to reconstruct each particle in three dimensions, followed by stereological quantification methods. The results, supported by small angle X-ray scattering characterization, reveal that SPIONs initially have a spherical shape, then grow increasingly asymmetric and irregular. A high heterogeneity in volume at the initial stages makes place for lower particle volume dispersity at later stages. The SPIONs settled into a preferred orientation on the support used for transmission electron microscopy imaging, which hides the extent of their anisotropic nature in the axial dimension, there by biasing the interpretation of standard 2D micrographs. This information could be feedback into the design of the chemical processes and the characterization strategies to improve the current applications of SPIONs in nanomedicine.

## 1. Introduction

Magnetosomes [1,2] are exceptional intracellular structures found in magnetotactic bacteria offering the cellular functionalities such as compasses for motility and orientation [3], oxygen chelation [4] or support in colonial self-assembly [5]. The core of magnetosomes consists of single-domain magnetite crystals [6], each possessing a magnetic moment that is thermally stable at physiological temperatures [7]. These magnetotactic crystals range from 30 nm to 120 nm in size [8,9]. The size and shape of the crystals can greatly affect their ability to perform their tasks and therefore the biomineralization is governed under precise control of a number of genetic factors [10,11,12]. Chemically produced superparamagnetic iron oxide nanoparticles (SPIONs) try to imitate the cores of these magnetosomes using magnetite (Fe_3_O_4_) [13] and maghemite (γ-Fe_2_O_3_) nanocrystals [14]. SPIONS can be produced at sizes ranging from roughly 10 nm to about 100 nm [15]. Their surface functionalization can be adjusted to generate biocompatibility [16] by means of well-described surface functionalization schemes [17,18].

SPIONs have been used in a broad portfolio of applications: in the oil industry [19], as chemical catalysts [20] or as effective separation technologies [21], but SPIONs are applied especially in biomedical technologies, owning to their biocompatibility and responsiveness to static and alternating magnetic fields. The global market for nanoparticles in biotechnology, drug development, and drug delivery has been estimated to have reached $79.8 billion in 2019, with a compound annual growth rate of 22% [22] and SPIONs play a significant role in this economical interest. For instance, as contrast enhancement agents for magnetic resonance imaging [23,24] and subsequent cell tracking [25] or as cell markers; for example, in stem cell applications [26]. Beyond the imaging applications, SPIONs have been suggested as drug delivery agents [27]. Reaching the target site is an issue many promising drug candidates face and advanced drug delivery using SPIONs may drastically influence modern medicine advancements [28]. Besides these complementary functions, magnetic hyperthermia is an emerging technology using SPIONs as a potential cancer treatment: after reaching the cancer cell the SPIONs are exposed to an alternating magnetic field (e.g., an MRI) and their response produces sufficient heat to destroy nearby cells [29,30,31].

As in the magnetosomes found in nature these applications rely on the accurate magnetization of the SPIONs, which depends significantly on their size and shape [32]. Shape control is one of the most exciting challenges in chemical nanotechnology [33,34], mainly because the interesting properties of nanoparticles can be tuned through variations in their morphology [35,36]. Additionally, shape can also affect their self-assembly behavior [37]. Although significant progress has been made, shape control of superparamagnetic oxides is still very challenging, in particular for small SPIONs, and syntheses that aim for shape control, except for few cases, such as cubes [38], and to a lesser extent nanorods [39], often result in rather broad particle size and shape distributions.

The introduction of shape anisotropy is particularly interesting in the context of magnetic fluid hyperthermia. It is one of the proposed strategies to increase the extent of heat generated by the particles without altering the chemical composition of the material used to fabricate the nanoparticles [40]. It is well established that shape anisotropy of nanoparticles leads to an enhancement of the dissipated heat since it adds to the magnetic anisotropy, and has a non-negligible effect on the time-dependent hysteresis of the particles. It has been shown, for example, that magnetite nanocubes [41] and nanorods [42,43] have a higher specific adsorption rate than spherical nanoparticles of similar size and made of the same material. While some investigations hint at the effect of sodium and potassium cations in driving the shape transition from spherical to cubic [44], the factors affecting the change in shape from spherical to ellipsoidal remain elusive, and might be related to the temperature profile imposed during the synthesis. For example, even though usually not explicitly mentioned in the literature, some of the high temperature organic syntheses of SPIONs lead to non-spherical nanoparticles, especially when larger sizes (about 20 nm diameter) are targeted.

Thus, a precise characterization of size and shape are extremely relevant for the biomedical functionality of SPIONs [45]. Yet typical size characterization techniques such as transmission electron microscopy [38], yields only two dimensional (2D) projections and assumptions on their form are needed to extract size. Indirect techniques (such as X-ray [46] and neutron [37] scattering) rely on the ensemble average of the particles, and thus, describe only an average anisotropy. Additionally, small-angle scattering techniques require mathematical models to describe particle shape [47]. Here, we synthesized SPIONs using the standard high-temperature decomposition method [48] and imaged the SPIONs using direct three dimensional (3D) imaging methods at consecutive time points during the thermal decomposition. A crucial aspect of this approach was the avoidance of shape models in the estimation of size (volume and surface area) and shape (sphericity and anisotropy) of each particle. To achieve this, we used geometrical techniques known as stereology [49]. Our results could be used to recognize bias in standard SPION size characterization methods and thereby help to improve SPIONs shape control, yielding better, more efficient materials for biomedicine.

## 2. Materials and Methods

### 2.1. SPIONs Synthesis

SPIONs were synthetized by thermally decomposing a previously synthetized iron oleate-complex using a modified literature procedure [48]. The iron oleate-complex was prepared by reacting iron chloride (FeCl_3_∙6H_2_O) with sodium oleate (equivalents 1:3). Then it was heated to 320 °C with a defined temperature ramp in presence of oleic acid in trioctylamine (98%) and kept at the final temperature for 35 min. During the synthesis 1 mL aliquots were extracted at different time points and quickly quenched. Then the resulting oleic acid coated SPIONs were separated by sequential centrifugations and re-dispersed in toluene.

### 2.2. Electron Tomography

The SPIONs suspensions were drop-casted onto single slot TEM supports (PlanoEM, Wetzlar, Germany), allowed to air-dry and introduced into a Tecnai spirit transmission electron microscope (ThermoFischer, Waltham, MA, USA) operated at 120 kV and equipped with a Veleta 2K camera (Olympus, Tokyo, Japan). The region of interest was chosen by systematic random sampling [50]: Three tilt series were recorded at stage coordinates [0,0], [500,0] and [−500,0]. The tilt series were recorded using serialEM software (Version 4.9.12) [51] ranging from −60° to +60° at 1° increment. The tilt series were reconstructed stacks in IMOD [52] using the particles as fiducial markers for alignment and weighted back projection [53] as reconstruction algorithm yielding tomograms (z-stacks) with isotropic voxels.

### 2.3. Image Processing

The particles were automatically selected from the tomograms using the “Analyze particles” algorithm in FIJI [54] and each particle was cropped into a subtomogram of dimensions 40 nm × 40 nm × original depth. In order to avoid bias, particles that were part of an aggregate—identified by testing if the particles touched the edges of the box—were excluded. After exclusion of the aggregates, an average of 102 subtomograms (+/−30) could be established per time point (a total of 512 particles), each containing the 3D data of a single particle. Renderings of the data were performed by the Volume Viewer plugin.

For the automated analysis in silico, binarization of the data was required. Therefore, the threshold for each subtomogram was set using the standard IsoData algorithm, followed by a watershed step. Finally, values outside the bounding box of the central particle were removed, which reduced noise. This yielded an average of 85 particles per time point (±35). From a 2D profile the equivalent spherical radius [55] was retrieved and the used to calculate the model-based volume. Inversely, model-free data could also yield the equivalent spherical radius: the radius of a particle with the same volume but shaped as a sphere.

For automated particle size measurements in 2D, the ‘measure particles’ tool of ImageJ (version 1.52u) was used. For the measurement in 3D, we used the 3D object counter plugin [56]. 

### 2.4. Stereology, Sphericity and Anisotropy

The ‘Grid’ function [57] of FIJI was used to place geometric probes digitally and randomly on the tomographic slices [58]. The crossings of the test lines, known as the Cavalieri estimator [59], were used for volume estimates (area per point = 0.91 nm^2^), and the vertical lines were used as the Fakir probe [60]. The ImageJ Reslice routine was used to obtain slices in the XZ and YZ orthogonal dimensions. Counting was done manually in ImageJ. For all calculations, the length density (spacing between the probes) was 0.95 nm in all dimensions. The estimated surface and volume were used to calculate sphericity using the following equation [61]:(1)$$\Psi =\frac{\sqrt[{3}]{36\times \pi \times V^{2}}}{A}$$
with *V* the stereologically estimated volume and *A* the stereologically estimated surface area of the particles.

The anisotropy was calculated using the following equation [62]:(2)$$Anisotropy=\sqrt{\frac{1}{2}}\;\frac{\sqrt{{\left(\lambda_{x}-\lambda_{y}\right)}^{2}+{\left(\lambda_{y}-\lambda_{z}\right)}^{2}+{\left(\lambda_{z}-\lambda_{x}\right)}^{2}}}{\sqrt{\lambda_{x}^{2}+\lambda_{y}^{2}+\lambda_{z}^{2}}}$$
with $\lambda_{x}$, $\lambda_{y}$, $\lambda_{z}$ the longest calipers at X, Y and Z angles. These quantities were retrieved by brute force computing using the transform plugin [63] of ImageJ. $\lambda_{x}$ (the maximum caliper length) was found by rotating the particle along the *x*-axis (0–180°) and measure the projection at each angle. Once found, the particle was rotated over this angle to align $\lambda_{x}$ in the XY plane. The Y angle, used to assess preferred orientations, was then measured using the ‘Measure particles…’ tool in ImageJ. The maximum width ($\lambda_{y}$) was found in a similar fashion by rotating 180° along the Z axis and again the particle was aligned in the XY plane. $\lambda_{z}$ was then the maximum particle width after rotating 90° along the Z axis. This was repeated for all particles.

### 2.5. Statistics

All statistical tests and graphs were performed in R (Version 3.6.1) [64]. A shapiro-Wilk test confirmed normality (*p* > 0.05) for all statistical datasets except the sphericity (Ψ) data. Therefore, the sphericity data is shown as boxplots, whereas all other data by arithmetic means and standard deviation. Comparisons were preceded by a homoscedasticity test to assure the variances were homogenous. This was done with Fisher’s f-test, using 5% as the significance level. If the Fischer’s f-test for homogenous variance resulted in a statistic *p* > 0.05, then it was assumed that both variances were homogenous, in which case a classic Student’s two-sample t-test was run to possible differences. If the Fischer’s f-test returned a *p* < 0.05, then a heteroscedastic situation was assumed (it was assumed that the variances of the two groups are different), in which case a Welch t-statistic was performed.

The kernel density algorithm disperses the mass of the empirical distribution function over a regular grid of at least 512 points and then uses the fast Fourier transform to convolve this approximation with a discretized version of the kernel and then uses linear approximation to evaluate the density at the specified points.

The difference of the means is calculated to compare the radii of gyration estimated from SAXS (mean ± confidence intervals) and from TEM (per particle data). If the lower bounds of the 95% confidence interval of the difference of the means was below zero (meaning 0 is within the confidence intervals), then no significant difference was assumed.

### 2.6. Small-Angle X-Ray Scattering

Small-angle X-ray scattering (SAXS) spectra were recorded using a NanoMax-IQ camera (Rigaku Innovative Technologies, Auburn Hills, MI, USA). The suspensions were kept in a 1 mm capillary at room temperature during the measurements. The raw data were processed with background and all possible artefacts taken into account [65] (a description of all data reduction steps and sequence can be found in Table 1 of this reference [66]). The scattering spectra are presented as a function of the momentum transfer:(3)$$q=\frac{4\pi \times \sin\left(\frac{\theta }{2}\right)}{\lambda }$$
where *θ* is the scattering angle and *λ* = 0.1524 nm is the photon wavelength. The radius of gyration $R_{g}$ was estimated in the Guinier regime, where a linear function was regressed against the linearized data defined by the natural log of the scattering intensity (I) against *q*^2^ [47]:(4)$$I\left(q\right)\propto \;e^{\frac{-R_{g}^{2}\cdot q^{2}}{3}}$$

Therefore, the natural log of the scattering intensity is proportional to $\frac{-R_{g}^{2}\cdot q^{2}}{3}$, which is in fact the slope defined by the low-q SAXS data and therefore the linear regression then estimates $R_{g}$.

## 3. Results

We used a modified version of Hyeon’s iron oleate based recipe [48] to synthesize monodisperse SPIONs with a target size of about 20 nm using a high boiling point solvent to control the particles size [67,68]. Aliquots were taken at five different time points during the thermal decomposition process (see Figure A1): the first one upon reaching the 320 °C plateau (=onset), then at three successive intervals of ten minutes starting ten minutes after onset and finally the fifth and last time point after cooldown of the particles. 

The samples were then characterized by transmission electron microscopy (TEM) and small angle X-ray scattering (SAXS). Tomographic tilt series [53,58] were acquired on the TEM ensuing the standard 2D profile but also 3D representation in silico of each single particle [27,69]. After tomographic reconstruction [52] at least 55 particles per time point were available for quantification. The time points in all figures are denoted by red minutes on yellow clock, with the blue circle denoting particles measured after cooling down the solution.

The particles appear as dark objects on a bright background in Figure 1. Each panel in Figure 1 shows a typical raw dataset at the designated time point. The orthogonally XY and YZ planes are shown: the position of the YZ plane is denoted by the white arrowheads in the XY plane. At each timepoint three particles were marked by yellow, violet and orange boxes in the XY plane: these particles were 3D rendered below each panel. They qualitatively show the difference in size and shape within this population of particles.

To characterize their size, two different quantification methods were used, both relying on the same dataset but with different assumptions and post-processing steps. Direct quantification of the entire particle’s volume without image processing was possible by the stereological quantification method known as the Cavalieri estimator [70,71] (the readers should consult reference [58] for an extended explanation, including examples). The alternative was the standard in the literature: automated analysis of 2D profiles in silico after a binarization step. This quantification assumes that the particle is spherical to estimate both its radius and volume.

### 3.1. Particle Volume

A wide variety of particle sizes could be observed at the onset of the 320 °C plateau (Figure 1A, Figure 2, Table 1). The majority of these particles had a volume below 2000 nm^3^. The mean volume was 1570 nm^3^ corresponding to a mean spherical radius of 6.86 nm (model-free) respectively 1921 nm^3^ and 7.11 nm (model-based). There were a few large outliers of more than 5000 nm^3^, which were the source of very high particle volume dispersity (Ð): 65.2%. 

The density kernel plots in Figure 2A are highly skewed and all outliers were larger than the mean sized particle. There was no statistically significant difference found between the two quantification methods.

Ten minutes later, the nanocrystals had increased almost eightfold in volume (Figure 1B, Table 1, Figure 2B): the mean volume of the particles was 12,085 nm^3^ (model-free) and 10,921 nm^3^ (model-based), a significant difference (*p* < 0.05). The model-free quantification result showed an equivalent spherical radius of 14.1, compared to 13.6 nm in the model-based result. The volume increase represents an average growth of 17.52 nm^3^ per second and the volume increase is highly significant compared to the first time point (*p* < 0.001). The smaller, younger cores catch up and larger ones grow slower due to a less favorable volume-to-surface ratio, yielding a decrease in polydispersity (but still rather high at 32.6%).

20 min after reaching the 320 °C plateau, the particles had further increased in volume (Figure 1C, Table 1, Figure 2C) but at much lower growth rates than previously measured (about 3.73 nm^3^ per second in average): the mean volume was 14,325 nm^3^ (model-free) and 13,133 nm^3^ (model-based), corresponding to an equivalent spherical radius of 15.0 nm and 14.5, respectively. Ð had dropped to 20.8%, the lowest since reaching the 320 °C plateau and these levels of variability were maintained for the rest of the reaction. The kernel density volume (Figure 2C) reveals near-symmetry: most particles had volumes within a non-significant range from the median, and outliers are found equally on both sides. Significant statistical differences were again found between the volume quantification of the model-based and model-free techniques (*p* < 0.05).

The system continued this close-to-linear growth of 3–4 nm^3^ per second in average and particle are always significantly (*p* < 0.001) larger than at the previous timepoint, except between 30′ and cooldown. After 30 min, a mean particle volume of 16,239 nm^3^ (model-free) and 13,979 nm^3^ (model-based) were reached, again a significant difference (*p* < 0.01). The corresponding to an equivalent spherical radius of 15.6 nm and 14.8, respectively.

After the reaction, the particles were allowed to cool down. A sample of the cooled down population showed slightly lower volumes in both quantification methods: mean volume of 15,324 (model-free) and 12,975 (model-based) or an equivalent spherical radius of 15.3 nm and 14.4 nm, respectively.

The model-based quantification systematically underestimated the volume and consequently the size of particles compared to the model-free quantification and the bias intensified in significance at later time points, including after cooling down.

The same particle batch was analyzed by small-angle X-ray scattering (Figure 3, Figure A2 and Figure A3). Only the radii of gyration $\left(R_{g}\right)\;$were retrieved to avoid biased of X-ray scattering analysis by the use of mathematical models for the particle shape. The radius of gyration is the orientationally averaged distance squared to the center of the mass of the particle and has the advantage that it is entirely model-free and can be retrieved both from the SAXS data and through image processing from the TEM data (Figure 3). There was no significant difference between the values of the radii of gyration estimated from SAXS and those estimated from TEM (Figure 3): the lower bound of the confidence interval of the difference of the means is always lower than 0 (the Δ values below each time point in Figure 3).

### 3.2. Sphericity, Anisotropy and Preferred Orientation

Shape variation can only be acquired using the full description from the entire 3D tomographic datasets of the object, not by making assumptions based on 2D profiles. Here, we use sphericity Ψ and the anisotropy to assess shape variation.

Sphericity is a unit-less measure between 0 and 1 that describes how closely an object resembles a sphere (See Material and Methods for the equation) [61]. Figure 4A shows the distribution of Ψ with a value of 1 matching a perfect sphere, shown by the dotted line. The median sphericity, $\tilde{\Psi }$, was the highest measured (0.92) at the onset and the 75% percentile was near 1 (0.99). Ten minutes later, $\tilde{\Psi }\;$had dropped significantly (*p* < 0.001) to 0.83. The particles stayed approximately at these values for the next ten minutes (see Table 1 for the mean values of Ψ). At time point 30 min at 320 °C, $\tilde{\Psi }$ had dropped further to 0.55, again significantly lower (*p* < 0.001) than ten minutes before. The values for $\tilde{\Psi }$ did not change significantly anymore during cooldown. This means that the particles started near-spherical but depart from that shape to become ellipsoidal. There are two significant jumps in sphericity: between onset and 10 min and between 20 min and 30 min.

The anisotropy data confirmed these observations. The anisotropy [62] describes the uniformity of the object and has magnitude of 0 for an isotropic body (a perfect sphere, shown as the dotted line in Figure 4B) but increases for anisotropic objects (see material and methods for equation). A significant (*p* < 0.05) difference in anisotropy magnitude was found between the first time point (onset of the 320 °C plateau) and ten minutes later, in accord with the sphericity data. Although noisier, the anisotropy follows the same trend as the $\tilde{\Psi }$: the particles start practically isotropic and grew increasingly anisotropic during the course of the reaction. The anisotropy was highest for the later stages and the cooled down particles, meaning these were the most anisotropic particles.

Two orthogonal axes were at all time points comparable in length: the difference between the largest caliper and the median caliper was not significant (Figure 4C). However, the third orthogonal axis, the shortest of the three orthogonal dimensions was significantly smaller (*p* < 0.001, red line in Figure 4C) and the difference increased with increasing time.

The direction of the maximum caliper was calculated for each particle (see Figure A4 for a graphical representation of the methodology) and represented by the two angles φ - rotation along the *x*-axis - and *θ* - rotation along the Y axis had an optimum at all time points in the broad vicinity of 90° (Figure 5A). Most significantly, after rotating each particle degrees (i.e., aligning the maximum anisotropy component in the XY orthogonal plane), a preferred orientation of the particles on the TEM grid was observed: there was an aversion for *θ* = 90° (the rotation in the Y-plane) but a preference peaked for *θ* at 69° and 111° (Figure 5B). The smaller particles at the onset (red in Figure 5) had the narrowest—rotation peak, which also aligned the closest to 90° in the rotation. The same was observed for the *θ*: the earlier stage had narrower peaks and was found closer to the 90° minimum. The entire scatter diagram can be consulted in Figure A5.

## 4. Discussion

### 4.1. The Thermal Decomposition Process

Unlike the controlled biomineralization of magnetotactic crystals in magnetosomes, the onset of the 320 °C plateau is marked by nucleation events, which is a stochastic process [72]. Particle nuclei spawn into existence at different time points with growth only limited by diffusion [35]. Particles are present at different stages of their growth as reflected by the very high volume polydispersity. Near-spheroidal shape is also a characteristic of this stage as the number of crystalline facets was still limited. This was seen by sphericity values near 1 and an anisotropy close to 0. During no other stage the particles attain such near-spherical shape. Indeed, the initial growth of these nuclei is well understood and growth occurs isotropically along distinctive facets [72,73]. When the nuclei reached the growth phase, their volume increased rapidly. This is marked by strong significant differences between the onset of the 320 °C plateau and 10 min later: volume, equivalent spherical radius and radius of gyration are all significantly different between these times points. Then, volume variability drops as larger particles suffer from less favorable volume-to-surface ratios. The increased surface area allowed for the adsorption of more molecules on the surface and this governs, together with the temperature, the free interfacial energy density that triggers dynamic changes in the nanocrystal shape [74]: the start of shape variation. Whereas the particles at onset of the 320 °C plateau were nearly spheroidal, morphologies appeared that can be described as octahaedral, coffin-like or bean-like in the 2D profiles but appear as an oblate spheroid when considered in 3D. This morphology is a reflection of the solvent-depending preferential binding of facets [35]. Dramatic changes in magnetic moments were observed 30 min after reaching the 320 °C plateau [75], which could be attributed to the increase of shape anisotropy. Indeed, our data show a considerable drop in sphericity and an increase in anisotropy at the same time point.

### 4.2. Measurement Disagreements

The disagreement between the two size quantification methods means either the model-free quantification overestimated the size and volume of the particles or the model-based quantification method underestimated those.

Using the radius of gyration, another model-free size quantification technique based on SAXS data agrees to the with the model-free TEM measurements. Hence, the assumption of spherical models was flawed. The mismatch between the two methods increased with increasing anisotropy: relying only on modelling of 2D profiles will lead to ambiguous interpretations when dealing with anisotropic particles.

However, tomographic reconstructions also have associated artefacts: the data acquisition covers a limited tilt angle range from −60° to +60°, resulting in missing information between −60° and −90° as well as 60° and 90°. This so-called missing wedge effect emerges as poor resolution in the XZ dimension after reconstruction, visible especially in the polar regions and apparent as a slight elongation. This missing wedge effect is not relevant in 2D profiles, but may play a role in the overestimation of the particle’s volume. Since the tomographic data is convoluted with the missing wedge effect the outcome of the quantification will be biased, i.e., overestimated due to the elongation along the Z axis. However, our data suggest that the missing wedge effect was not significantly distorting the interpretations in this setup because (A) all particles were recorded using the exact same tilting protocol, which produces the same missing wedge effect strength in all particles. Hence, the missing wedge alone cannot explain the observed increase in anisotropy; (B) the model-free TEM data was confirmed by SAXS data at all time points; (C) more spherical particles (at the onset of 320 °C) suffered less from the preferred orientation than more anisotropic particles observed at later stages as shown by the narrower peaks and the alignment with the 90° angle. (D) the discrepancy between the model-based method and model-free method grows with increasing size. Therefore, we consider the impact of the missing wedge effect not to be significant on the relative trend over the measured time points, although it may shift the absolute values of the anisotropy and sphericity measures.

### 4.3. Model-Based Versus Model-Free Quantifcation

Our data demonstrates the inaccuracies of model-based quantification methods such as conventional TEM to characterize anisotropic nanoparticles. The source of the problem of model-dependent interpretations is the heterogeneous shape of the particles in its three dimensions. Thus, assuming a single shape-model is not appropriate to properly describe the ensemble of these populations. Conventional transmission electron microscopy is single-particle based and can portray a multitude of shapes in a mixture but produces a 2D projection of each 3D object, thereby neglecting the entire axial dimension. This again creates the need for model-based interpretations and may lead to biased size analysis. TEM, combined with tomography, is a procedure that reconstructs anisotropic objects in their three dimensions in silico and with isotropic voxels at sub-nanometer resolution [27,76] Based on these data, we found that SPIONs display a preferred orientation on the TEM support film: around 20–25° tilted at the longest axis. It is unclear why exactly these positions are preferred, but referred orientations are a known and a delicate topic in single particle reconstruction [77,78,79]. The volume of an oblate spheroid tilted along such angles would be partially hidden in a 2D projection, which leads to an underestimation of the volume by a 6–9% (about upon assuming a spherical shape. Indeed, a correction factor of 7% nullifies all significant differences between the volume quantification methods for these particular objects.

On the one hand, this means that this bias in the TEM characterization is not of concern for most biomedical SPIONs related applications. A possible exception would be applications of heating ferrofluid to achieve hyperthermia [80], where it was shown that an error of 5% can half the temperature rise rate in magnetite nanoparticles [81]. On the other hand, there are nanotechnological applications, such as plasmonic sensing, where the error in precisely estimating the size and the shape is truly unforgiving [58,82].

## 5. Conclusions

In conclusion, we assessed how the accuracy of size and shape characterization of SPIONs by transmission electron microscopy. To that extend, we applied 3D TEM imaging techniques to fully describe, i.e., in all three dimensions, growing SPIONs during the thermal decomposition reaction. The heterogeneity of the samples could be statistically portrayed and from the analysis, it could be observed that SPIONs are almost spherical at onset but then grow increasingly more anisotropic in shape. The preferred orientation of the objects on TEM support films renders spherical shape assumptions void but the effect is insufficient to currently perturb the structure-function relationship of SPIONs.

## Figures and Tables

**Figure 1 materials-13-02018-f001:**
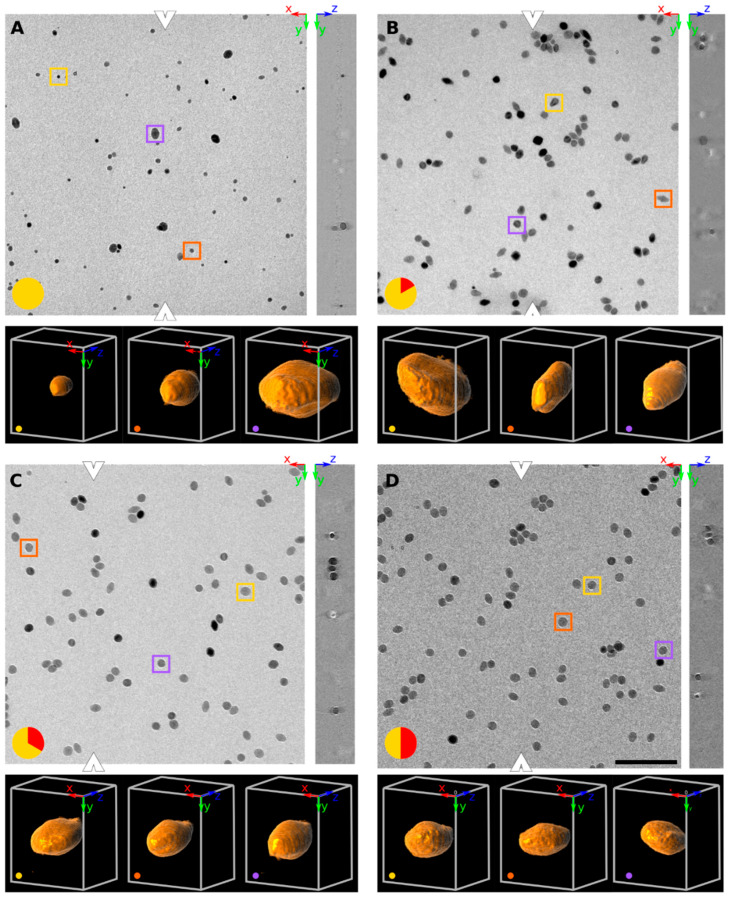
Orthogonal slices (XY and YZ plane) of SPIONs at four different time points at 320 °C. At each time point, three particles were selected (yellow, orange and purple boxes) and 3D rendered below each panel (at a viewpoint of [X, Y, Z] angle of 30°, −10° and 90°). The time in minutes since reaching the 320 °C plateau is encoded in red on a yellow disk at the bottom left of each time point (**A**: reaching 320 °C, **B** after 10′, **C** after 20′ **D** after 30′). Representative tomograms are shown in XY and YZ planes of the aliquot taken. The white arrowheads denote where the YZ plane was sampled. The scale bar denotes 200 nm and all images (both in XY and YZ plane) have the same magnification and resolution.

**Figure 2 materials-13-02018-f002:**
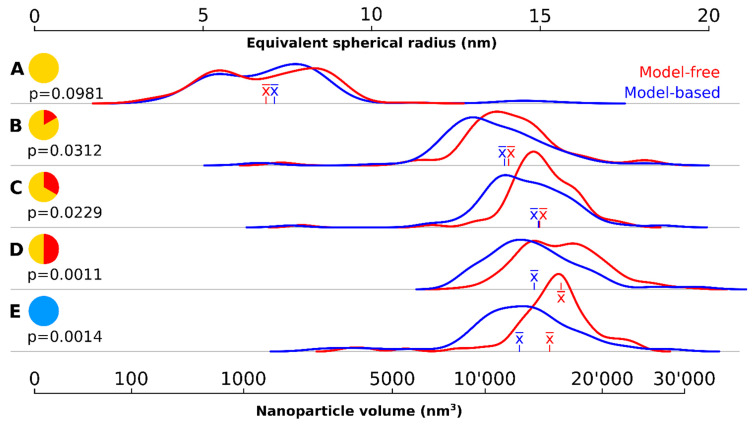
Density kernel plots of the particle size at different time points using two different quantification methods. (**A**) at the onset of the 320 °C plateau; (**B**) after 10 min at 320 °C; (**C**) after 20 min at 320 °C; (**D**) after 30 min at 320 °C; (**E**) after cool down to room temperature. The time is symbolized on the left of each time point. The red curves show the particle volume distribution using the model-free, stereological volume estimator. The blue curves correspond to model-based calculations of the equivalent spherical radius using 2D image profiles (assuming a spherical particle). The mean ($\bar{x}$) is shown in the respective color of the curve for each time point. The p values correspond to a Student’s t-test statistic describing the differences between the two curves at each time point.

**Figure 3 materials-13-02018-f003:**
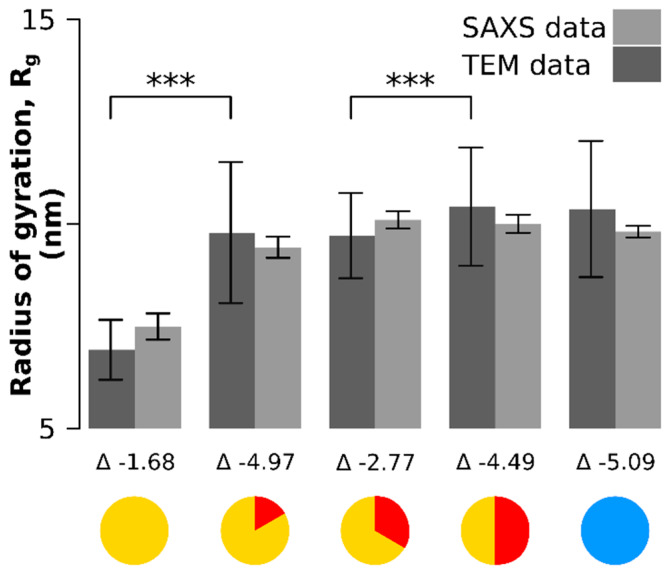
Comparison of the radii of gyration between SAXS and TEM measurements (model-free data). The lower bound 95% confidence interval of the difference between the means (denoted as Δ) is always negative, which means there is no significant differences between the SAXS and TEM measurements. Significant differences between time points in radius of gyration were observed between the onset time point and the 10 min time point and between the 20 min and 30 min time point. The larger error in the TEM-based quantification originates from smaller sample sizes and a different counting mode (TEM counting and SAXS fit). *** means *p* < 0.001.

**Figure 4 materials-13-02018-f004:**
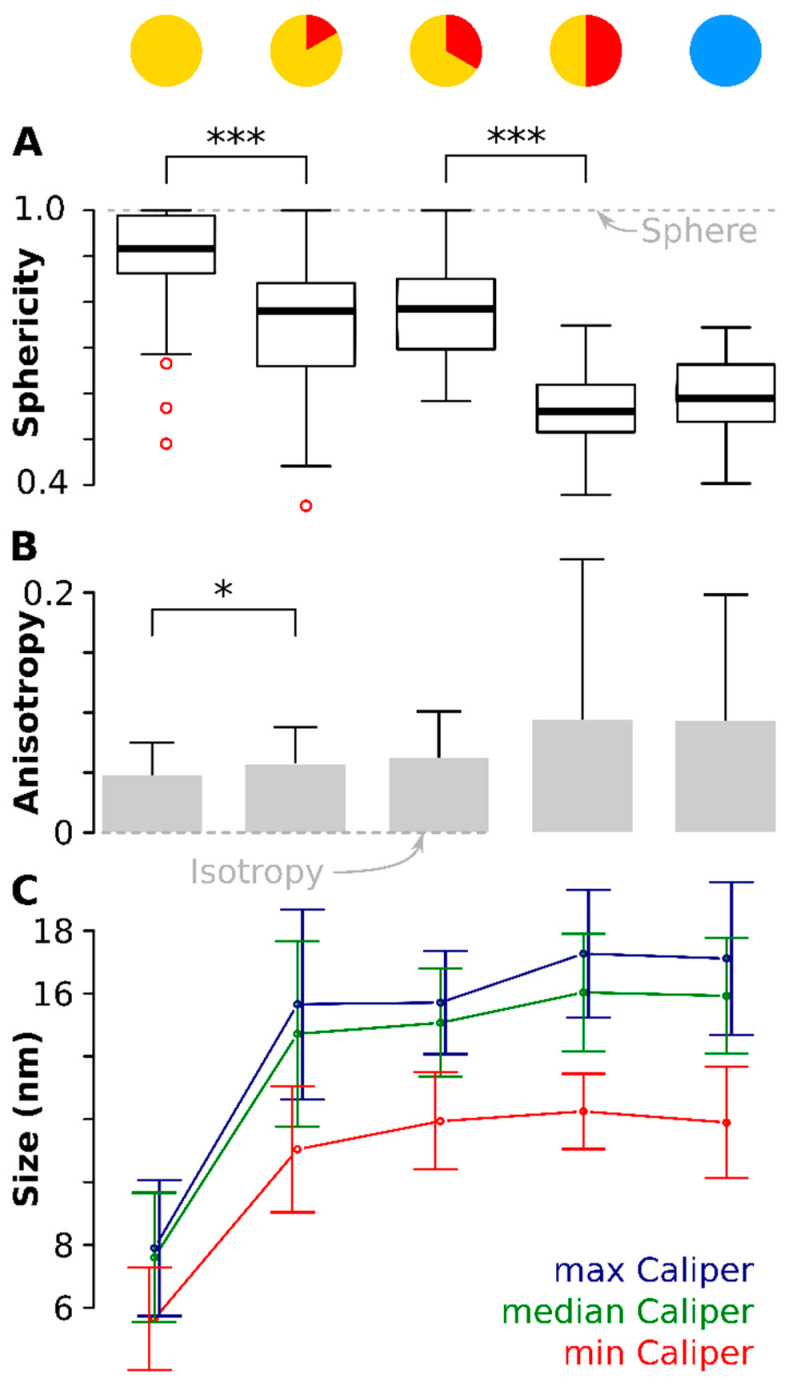
Particle anisotropy during the thermal decomposition process. (**A**) shows boxplots (because of the non-normal distribution) of the sphericity (1 = sphere, denoted by the dotted gray line). There is a significant drop (*p* < 0.001) in sphericity of the particles between onset of the 320 °C plateau and again ten minutes later. A second drop (again *p* < 0.001) is observed between 20 min and 30 min after onset of the 320 °C plateau. The red circles denote outliers. (**B**) shows the anisotropy of the particles at different time points (0 = isotropy, denoted by the dotted gray line). Again, a significant difference (*p* < 0.05) is seen between the situation at the onset of the 320 °C plateau and ten minutes later. (**C**) shows the development of the three orthogonal caliper lengths during the thermal decomposition. Plotted is the mean caliper of the longest, median and shortest axis. Note: for readability, the error bars are slightly shifted. * means *p* < 0.05; *** means *p* < 0.001.

**Figure 5 materials-13-02018-f005:**
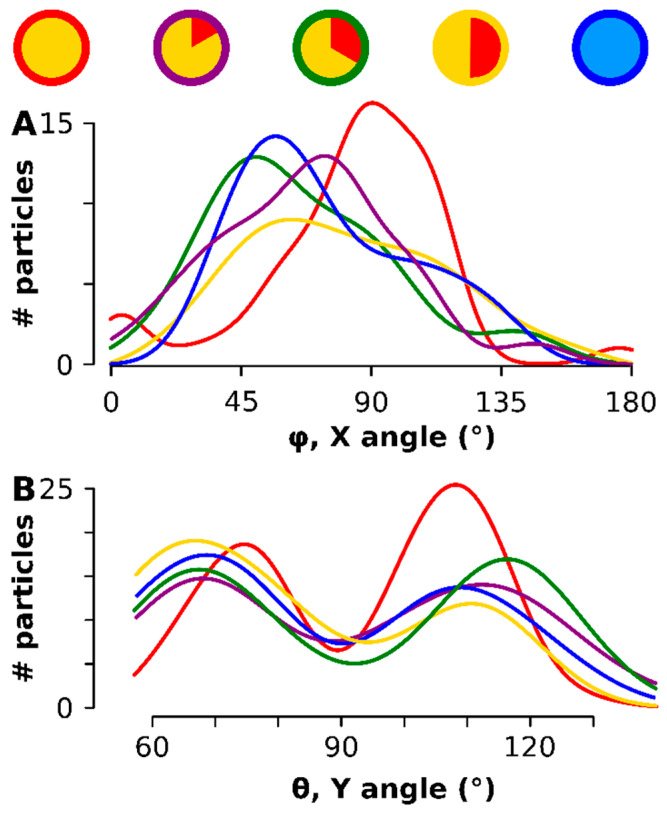
The 3D orientation of each particle on the TEM support. The time points are color coded as shown on top. (**A**) shows the orientation of the longest axis, i.e., angle φ the rotation along the *X*-axis- (**B**) shows the orientation of *θ*, the rotation around the *Y*-axis.

**Table 1 materials-13-02018-t001:** Summary of the analysis of the tomographic 3D datasets. Means are provided as average ± standard deviation.

Measure	Onset	10′	20′	30′	Cooldown
**Particle count**	149	112	83	96	72
**Volume (nm^3^)**					
Mean	1597 ± 1040	12,085 ± 3938	14,325 ± 2981	16,239 ± 3849	15,324 ± 3732
Median	1470	11,341	13,990	16,052	15,400
**Surface area (nm^2^)**					
Mean	679 ± 287	3528 ± 1471	3746 ± 927	7700 ± 1766	5144 ± 1490
Median	642	3124	3559	5383	5019
**Shape Factors**					
Sphericity	0.91 ± 0.09	0.76 ± 0.12	0.77 ± 0.10	0.56 ± 0.09	0.59 ± 0.08
Anisotropy	0.047 ± 0.027	0.058 ± 0.031	0.062 ± 0.039	0.0831 ± 0.08	0.082 ± 0.040

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
