# Peer review of "Characterization of the Shape Anisotropy of Superparamagnetic Iron Oxide Nanoparticles during Thermal Decomposition"

_materials, 2020, doi:10.3390/ma13092018_

Round 1

Reviewer 1 Report

The manuscript characterizes and study the shape anisotropy of iron oxide nanoparticles. The TEM is majorly utilized to characterize.

In general, the manuscript is well written and presented. Language is easy to understand. However, I feel that manuscript lacks the study on material application, which can increase the readership of this study.  

Objectives are clear.

Minor

I found the latest review article, which I think can be cited in the manuscript. https://www.mdpi.com/1422-0067/21/7/2455/htm

Include scale bar for all images in figures.

Author Response

Reviewer 1

In general, the manuscript is well written and presented. Language is easy to understand. However, I feel that manuscript lacks the study on material application, which can increase the readership of this study. I found the latest review article, which I think can be cited in the manuscript. https://www.mdpi.com/1422-0067/21/7/2455/htm

We thank the reviewer for this remark. We suggest to expand the introduction on the applications of SPIONs. We changed and considerably extended paragraph 2 of the introduction to:
SPIONs have been used in a broad portfolio of applications: in the oil industry[1], as chemical catalysts[2] or as effective separation technologies[3]. But SPIONs are applied especially in biomedical technologies, owning to their biocompatibility and responsiveness to static and alternating magnetic fields. The global market for nanoparticles in biotechnology, drug development, and drug delivery has been estimated to have reached $79.8 billion in 2019, with a compound annual growth rate of 22%[4] and SPIONs play a significant role in this economical interest. For instance as contrast enhancement agents for magnetic resonance imaging[5,6] and subsequent cell tracking[7] or as cell markers; for example for stem cells[8]. Beyond the imaging applications, SPIONs have been suggested as drug delivery agents [9]. Reaching the target site is an issue many promising drug candidates face and advanced drug delivery using SPIONs may drastically influence modern medicine advancements[10]. Besides these complementary functions, magnetic hyperthermia is an emerging technology using SPIONs as a potential cancer treatment: after reaching the cancer cell the SPIONs are brought into an alternating magnetic field (e.g. an MRI) and their response produces sufficient heat to destroy nearby cells[11–13].
As in the magnetosomes found in nature these applications rely on the accurate magnetization of the SPIONs, which depends significantly on their size and shape[14].

The suggested citation was added to the manuscript.

Include scale bar for all images in figures.

The authors thank the reviewer for reminding us. We checked all scale bars. Given all images in Figure 1 are the same magnification and resolution and for reasons of readability of the image we did not put the same scale bar on all 8 images in the figure. In Figure 1 we added a sentence to the caption that makes clear that the scale bar in Figure 1D is valid for all images in the figure.

References used in this reply

1. Ko, S.; Huh, C. Use of nanoparticles for oil production applications. Journal of Petroleum Science and Engineering 2019, 172, 97–114, doi:10.1016/j.petrol.2018.09.051.
2. Manaenkov O. V.; Matveeva V. G.; Sinitzyna P. V.; Ratkevich E. A.; Kislitza O. V.; Doluda V. Y.; Sulman E. M.; Sidorov A. I.; Mann J. J.; Losovyj Y.; et al. Magnetically recoverable catalysts for cellulose conversion into glycols. Chemical Engineering Transactions 2016, 52, 637–642, doi:10.3303/CET1652107.
3. Digigow, R.G.; Dechézelles, J.-F.; Kaufmann, J.; Vanhecke, D.; Knapp, H.; Lattuada, M.; Rothen-Rutishauser, B.; Petri-Fink, A. Magnetic microreactors for efficient and reliable magnetic nanoparticle surface functionalization. Lab Chip 2014, 14, 2276–2286, doi:10.1039/C4LC00229F.
4. Nano-safety: what we need to know to protect workers; Fazarro, D., Ed.; De Gruyter textbook; De Gruyter: Berlin ; Boston, 2017; ISBN 978-3-11-037375-2.
5. Stephen, Z.R.; Kievit, F.M.; Zhang, M. Magnetite nanoparticles for medical MR imaging. Materials Today 2011, 14, 330–338, doi:10.1016/S1369-7021(11)70163-8.
6. Dadfar, S.M.; Camozzi, D.; Darguzyte, M.; Roemhild, K.; Varvarà, P.; Metselaar, J.; Banala, S.; Straub, M.; Güvener, N.; Engelmann, U.; et al. Size-isolation of superparamagnetic iron oxide nanoparticles improves MRI, MPI and hyperthermia performance. J Nanobiotechnol 2020, 18, 22, doi:10.1186/s12951-020-0580-1.
7. Barrow, M.; Taylor, A.; Fuentes-Caparrós, A.M.; Sharkey, J.; Daniels, L.M.; Mandal, P.; Park, B.K.; Murray, P.; Rosseinsky, M.J.; Adams, D.J. SPIONs for cell labelling and tracking using MRI: magnetite or maghemite? Biomater. Sci. 2018, 6, 101–106, doi:10.1039/C7BM00515F.
8. Jasmin, .; Souza, G.T.; Andrade Louzada, R.; Rosado-de-Castro, P.H.; Mendez-Otero, R.; Carvalho, A.C.C. Tracking stem cells with superparamagnetic iron oxide nanoparticles: perspectives and considerations. IJN 2017, Volume 12, 779–793, doi:10.2147/IJN.S126530.
9. Bonnaud, C.; Monnier, C.A.; Demurtas, D.; Jud, C.; Vanhecke, D.; Montet, X.; Hovius, R.; Lattuada, M.; Rothen-Rutishauser, B.; Petri-Fink, A. Insertion of Nanoparticle Clusters into Vesicle Bilayers. ACS Nano 2014, 8, 3451–3460, doi:10.1021/nn406349z.
10. Laurent, S.; Saei, A.A.; Behzadi, S.; Panahifar, A.; Mahmoudi, M. Superparamagnetic iron oxide nanoparticles for delivery of therapeutic agents: opportunities and challenges. Expert Opinion on Drug Delivery 2014, 11, 1449–1470, doi:10.1517/17425247.2014.924501.
11. Laurent, S.; Dutz, S.; Häfeli, U.O.; Mahmoudi, M. Magnetic fluid hyperthermia: Focus on superparamagnetic iron oxide nanoparticles. Advances in Colloid and Interface Science 2011, 166, 8–23, doi:10.1016/j.cis.2011.04.003.
12. Gamarra, L.; Silva, A.C.; Oliveira, T.R.; J. B. Mamani; Malheiros, S.M.F.; Malavolta, L.; Pavon, L.F.; Sibov, T.T.; Amaro Jr, E.; Gamarra, L. Application of hyperthermia induced by superparamagnetic iron oxide nanoparticles in glioma treatment. IJN 2011, 591, doi:10.2147/IJN.S14737.
13. Piazza, R.D.; Viali, W.R.; dos Santos, C.C.; Nunes, E.S.; Marques, R.F.C.; Morais, P.C.; da Silva, S.W.; Coaquira, J.A.H.; Jafelicci, M. PEGlatyon-SPION surface functionalization with folic acid for magnetic hyperthermia applications. Mater. Res. Express 2020, 7, 015078, doi:10.1088/2053-1591/ab6700.
14. de Montferrand, C.; Hu, L.; Milosevic, I.; Russier, V.; Bonnin, D.; Motte, L.; Brioude, A.; Lalatonne, Y. Iron oxide nanoparticles with sizes, shapes and compositions resulting in different magnetization signatures as potential labels for multiparametric detection. Acta Biomaterialia 2013, 9, 6150–6157, doi:10.1016/j.actbio.2012.11.025.

Reviewer 2 Report

The manuscript reports on the synthesis of superparamagnetic iron oxide nanoparticles (SPIONs) and on the study of their microstructure and morphological properties. The SPIONs were prepared by thermally decomposing a previously synthetized iron oleate-complex. Transmission electron microscopy (TEM) and small angle X-ray scattering were measured as a function of time during the annealing step. Additionally, the TEM measurements also involved tilting to obtain the Z axis dimension, for 3D particle reconstruction. The objective is to develop a model free procedure to ascertain the evolution of the nanoparticles dimensions, with the heat treatment process, and to understand its implications in the developments of SPIONs base treatments. The manuscript has original results, but needs minor revisions. I have the following comments:

- On page 4 it is written that “Raw data were processed according to standard procedures …”. It would be better to give a reference there, to know what these standard procedures are.

- On page 6 it is written that “The smaller, younger cores catch up and larger ones grow slower due to a less favorable volume-to-surface ratio yielding a decrease in polydispersity (but still rather high at 32.6 %). was significantly lower (p<0.05) than the model-free assessments (1st and 2nd boxplots).”. Part of the second sentence seems to be missing. What were the authors intending to say with that sentence ?

- Several times, the text refers to a figure 3B. However, the manuscript only presents figure 3 without separation between 3A or 3B, which is confusing. Nevertheless, what the authors refer in the text as 3B seems to also be in figure 3. It would be better to clarify this.

- On Page 7 it is written that “In order to avoid an analysis of the scattering data biased by the use of mathematical models for the particle shape, only and the radii of gyration were retrieved”. There seems to be missing something after the “only”. How was it retrieved from the SAXS data ?

- The authors start by referring to magnetotatic bacteria, although the study itself deals with chemically produced SPIONs. Would the present study be relevant to study the morphological properties of the bacteria (for example, in vivo) ? Can the authors comment on this.

Author Response

Reviewer 2

On page 4 it is written that “Raw data were processed according to standard procedures …”. It would be better to give a reference there, to know what these standard procedures are.

Thank you for pointing this out. The sentence is indeed quite vague. We hope we could sufficiently improve it when we change the sentence to:
The raw data were processed with background and all possible artefacts taken into account [1] (a description of all data reduction steps and sequence can be found in Table 1 of this reference [2]).
Two new references were added:
Grillo, I. Small-Angle Neutron Scattering and Applications in Soft Condensed Matter. In Soft Matter Characterization; Borsali, R., Pecora, R., Eds.; Springer Netherlands: Dordrecht, 2008; pp. 723–782 ISBN 978-1-4020-4464-9.
Pauw, B.R. Everything SAXS: small-angle scattering pattern collection and correction. J. Phys.: Condens. Matter 2013, 25, 383201, doi:10.1088/0953-8984/25/38/383201.

On page 6 it is written that “The smaller, younger cores catch up and larger ones grow slower due to a less favorable volume-to-surface ratio yielding a decrease in polydispersity (but still rather high at 32.6 %). was significantly lower (p<0.05) than the model-free assessments (1st and 2nd boxplots).”. Part of the second sentence seems to be missing. What were the authors intending to say with that sentence ?

We thank the reviewer for showing us this mistake. Clearly, a part of an older sentence was not completely deleted. We deleted the partial sentence to make:
“The smaller, younger cores catch up and larger ones grow slower due to a less favorable volume-to-surface ratio yielding a decrease in polydispersity (but still rather high at 32.6 %).

Several times, the text refers to a figure 3B. However, the manuscript only presents figure 3 without separation between 3A or 3B, which is confusing. Nevertheless, what the authors refer in the text as 3B seems to also be in figure 3. It would be better to clarify this.

We thank the reviewer for pointing this out. Again, this is clearly a mistake which originated when we decided to move the second part of Figure 3 to the supplemental information (Figure 3B has become Figure SI-2). We cleared up the references to this figure panel and now the reference ‘Figure 3B’ is not anymore found in the manuscript.

On Page 7 it is written that “In order to avoid an analysis of the scattering data biased by the use of mathematical models for the particle shape, only and the radii of gyration were retrieved”. There seems to be missing something after the “only”. How was it retrieved from the SAXS data ?

We thank the reviewer for pointing this out. Indeed, this part is quite poorly written. Therefore, we rewrote the entire paragraph:
The same particle batch was analyzed by small-angle X-ray scattering (Figure 3 and Figure SI-2). Only the radii of gyration were retrieved to avoid biased of X-ray scattering analysis by the use of mathematical models for the particle shape. The radius of gyration has the advantage that it is entirely model-free and can be retrieved both from the SAXS data and through image processing from the TEM data (Figure 3). There was no significant difference between the radii of gyration performed on SAXS and the radii of gyration performed on TEM (Figure 3): the lower bound of the confidence interval of the difference of the means is always lower than 0 (the Δ values below each time point in Figure 3).

The authors start by referring to magnetotatic bacteria, although the study itself deals with chemically produced SPIONs. Would the present study be relevant to study the morphological properties of the bacteria (for example, in vivo) ? Can the authors comment on this.

We thank the reviewer for this question. Magnetosomes in bacteria occur in chains and magnetosome genes encode for proteins control the crystallographic signature of the magnetite crystals (the size and morphology) in magnetotactic bacteria, as well as magnetosome chain organization. The technique of electron tomography, as applied in our manuscript, has been used to study the organization of these magnetosome chains in situ[3]. However, the presence of biological material requires to perform the electron tomography at low temperatures to avoid structural loss. The technique is then known as cryoelectron tomography, which yielded a Nobel price in 2017. A comprehensive review of prokaryotic cryoelectron tomography is given here: [18].
In order to keep the manuscript concise, we decided not to take the discussion in this direction. The authors hope the reviewer understands this decision.

References used in the reply

1. Grillo, I. Small-Angle Neutron Scattering and Applications in Soft Condensed Matter. In Soft Matter Characterization; Borsali, R., Pecora, R., Eds.; Springer Netherlands: Dordrecht, 2008; pp. 723–782 ISBN 978-1-4020-4464-9.
2. Pauw, B.R. Everything SAXS: small-angle scattering pattern collection and correction. J. Phys.: Condens. Matter 2013, 25, 383201, doi:10.1088/0953-8984/25/38/383201.
3. Abreu, F.; Sousa, A.A.; Aronova, M.A.; Kim, Y.; Cox, D.; Leapman, R.D.; Andrade, L.R.; Kachar, B.; Bazylinski, D.A.; Lins, U. Cryo-electron tomography of the magnetotactic vibrio Magnetovibrio blakemorei: Insights into the biomineralization of prismatic magnetosomes. Journal of Structural Biology 2013, 181, 162–168, doi:10.1016/j.jsb.2012.12.002.
4. Oikonomou, C.M.; Chang, Y.-W.; Jensen, G.J. A new view into prokaryotic cell biology from electron cryotomography. Nat Rev Microbiol 2016, 14, 205–220, doi:10.1038/nrmicro.2016.7.